# Effect of Amnioguard and Biomesh GTR Membranes with Novabone Putty in the Treatment of Periodontal Osseous Defects—A Communication

**DOI:** 10.3390/ijerph20010816

**Published:** 2023-01-01

**Authors:** Priyankar Chakraborthy, Potluri Leela Ravishankar, Anbukumari Vadivelu Saravanan, Khalid J. Alzahrani, Ibrahim F. Halawan, Saleh Alshammeri, Mrim M. Alnfiai, Hosam Ali Baeshen, Sarah Yasser M. Qattan, Ammar Almarghlani, Sunanda Rao Karkala, Anila Neelakandan, Shankargouda Patil

**Affiliations:** 1Department of Periodontology, SRM Kattankulathur Dental College & Hospital, SRM Institute of Science and Technology SRM Nagar, Kattankulathur 603203, India; 2Department of Clinical Laboratories Sciences, College of Applied Medical Sciences, Taif University, Taif 21944, Saudi Arabia; 3Department of Optometry, College of Applied Medical Sciences, Qassim University, Buraydah 52571, Saudi Arabia; 4Department of Information Technology, College of Computers and Information Technology, Taif University, Taif 21944, Saudi Arabia; 5Department of Orthodontics, College of Dentistry, King Abdulaziz University, Jeddah 21589, Saudi Arabia; 6Private Practice, Jeddah 21586, Saudi Arabia; 7Department of Periodontics, Faculty of Dentistry, King Abdulaziz University, Jeddah 21589, Saudi Arabia; 8College of Dental Medicine, Roseman University of Health Sciences, South Jordan, UT 84095, USA

**Keywords:** AmnioGuard, BioMesh, NovaBone putty, periodontal regeneration

## Abstract

(1) Background: Guided tissue regeneration was an effective surgical procedure in the management of intrabony defects and has undergone a number of changes in terms of materials and techniques. The aim of this study is to compare AmnioGuard and BioMesh in combination with NovaBone putty in intrabony defects. (2) Methodology: Ten patients who needed regenerative periodontal therapy were randomly allocated into two groups based on the inclusion criteria. These patients were subjected to phase I therapy followed by which Group A patients were treated with AmnioGuard + NovaBone putty whereas Group B with BioMesh + NovaBone putty. The clinical indices were obtained at baseline, 3 months and 6 months post-operatively while radiographic parameters were obtained at 6 months post-op. (3) Results & Conclusion: At six months after surgery, Group B (33% bone gain) showed a statistically significant change from Group A (16% bone gain) in both the clinical and radiographic measures (*p* < 0.05).

## 1. Introduction

Periodontitis is a periodontal disease that causes irreparable loss of connective tissue attachment and supporting alveolar bone. These modifications frequently result in aesthetically and functionally impaired dentition. Periodontists have been interested in rebuilding tissues lost by periodontitis in bone defects for many decades. Periodontal regeneration is the complete return of the missing tissues to their original form and function by reenacting the key wound-healing processes that were involved in their formation.

Isolation for selective cell repopulation has been a recognized therapeutic approach for treating periodontal intrabony abnormalities for almost 30 years. Melcher [1] originally explored compartmentalizing tissues to promote regeneration in 1976, and Nyman et al. were the first to show how this technique could be used in practice [2]. Gottlow et al. [3], created the name guided tissue regeneration (GTR) in 1986 to characterize this therapy approach, which permits bone, cementum, and periodontal ligament to regenerate in degranulated osseous defects. The use of GTR therapy has varied and evolved since it was first used. Initially, degranulated periodontal infrabony abnormalities were just isolated using non-resorbable occlusal barriers. Over time, defects were filled with bone and bone substitutes, and when a barrier was placed over the defect, the term “combination GTR” was adopted. After some time, bioresorbable barriers took the role of non-resorbable barriers, and more recently, biological growth factors have been utilized to speed up healing. The GTR procedure is frequently used in conjunction with the implantation of bone grafts or bone graft replacements beneath the membrane. These grafts sustain the barrier material and keep it from collapsing into the defect or onto the root surface.

In recent years, biological barrier membranes have gained importance in periodontal regeneration. Amniotic membrane (AmnioGuard^®^-Biocover laboratories, Haryana, India which was first used in skin transplantation in 1910, has now been used in the reconstruction of the oral cavity [4]. Another membrane in this field is a combination of polylactide and polyglycolide namely BioMesh^®^ (Samyang Corporation, Korea). The interconnective porous structure promotes blood vessel formation, cell occlusiveness and biodegradability. NovaBone Putty^®^ (NB), Bangalore, India a bioactive glass, is a synthetic alloplastic biocompatible material that has both osteostimulative and osteoconductive characteristics that has been used as a bone replacement graft for periodontal osseous defects. It is a new generation calcium phospho-silicate bone graft material additives like polyethylene glycol and glycerol to improve handling and efficacy. The use of NB as a bone graft is a viable option in bone regenerative strategies. The study aims to compare AmnioGuard^®^ and BioMesh^®^ (Haryana, India) in the management of intrabony deficiencies along with NovaBone^®^ Putty (Bangalore, India).

## 2. Materials and Methods

### 2.1. Study Design

The study is a double-blinded, randomized, prospective, split-mouth study. A total of 10 subjects visiting the department of Periodontics, SRM Kattankulathur Dental College and Hospital, presenting with generalized stage III grade B or C periodontitis (AAP 2017) having one or more vertical osseous defects were enrolled in the study. Patients with good oral hygiene compliance, at least 2 periodontal pockets with probing depth (≥6 mm) on contralateral side of the same arch or on the opposite arch after Phase I Therapy and at least 2 intrabony defects in interproximal areas with radiographic evidence of intrabony compartment (≥4 mm) on contralateral side of the same arch or on the opposite arch were included as part of the study. Patients with systemic diseases that might influence periodontal conditions, pregnant/ lactating mothers and those who received any periodontal treatment and medication for the past six months were excluded from the study. All the patients were duly informed about the procedure and informed consent was obtained. The study was approved by the Institutional Ethical Committee of SRM Dental College, Chennai. (Ethical Approval number: 895/IEC/2015). The bilateral sites were randomly assigned into two groups, Group A—AmnioGuard^®^ and NovaBone^®^ Putty and Group B—BioMesh^®^ and NovaBone^®^ Putty. 

### 2.2. Clinical Recordings

The clinical recordings were evaluated pre-operatively and at 3 and 6 months post-surgically using acrylic stents individually fabricated for the patients. Relative attachment level (RAL) [5] and Pocket probing depth (PPD) [6] were recorded with the help of William’s periodontal probe, from the occlusal level of the acrylic stent to the base of the periodontal pocket and from the base of the periodontal pocket to the crest of the gingival margin, respectively.

### 2.3. Radiographic Assessment

Radiovisiography imaging was used to assess the sites at baseline, 3 and 6 months (Figure 1, Figure 2, Figure 3, Figure 4, Figure 5 and Figure 6) using paralleling technique and sensor holders. For measurements, the connector line tool of the software was used. Defects were radiographically evaluated to measure average bone fill and percentage of bone fill. All the observations were recorded and subjected to statistical analysis. 

### 2.4. Surgical Procedure

Following Phase I therapy, the patients were reevaluated to confirm the inclusion criteria. Extraoral and intraoral areas were disinfected with 5% povidone iodine and 0.2% chlorhexidine mouthwash, respectively. The interdental papilla was retained while a full-thickness mucoperiosteal flap was reflected after the surgical site had been anesthetized with 2% Xylocaine HCl and adrenaline (1:80,000). After flap reflection and exposure of the osseous defect, a thorough debridement was done followed by irrigation with Normal saline. Defect depth was measured from the bottom of the defect to the clinically visible cemento-enamel junction using William’s probe and graft placed after pre-suturing.

Randomly selected intrabony defect sites were covered with the AmnioGuard [7,8,9] and BioMesh membranes, respectively. Both membranes were selectively trimmed according to the defect area and placed. Interrupted sutures were placed to obtain the primary closure of the interdental papillae and the area was protected with Coe-Pak. Systemic antibiotics and analgesics were prescribed. Post-operative instructions were given to all the patients.

### 2.5. Statistical Analysis

IBM SPSS statistics software 23.0 was used to analyze the data that was gathered. Frequency analysis, percentage analysis, mean and standard deviation were employed for categorical variables and continuous variables, respectively, to describe the data using descriptive statistics. The Mann–Whitney U test was used to determine whether there was a significant difference between the bivariate samples in Independent groups. The Friedman test and Wilcoxon signed rank test was applied for the multivariate analysis in repeated measures. The probability value of 0.05 is regarded as a significant level in all of the aforementioned statistical techniques.

## 3. Results

Out of the 10 patients, 6 were male and 4 were female. The age group ranged from 25–45 years. At three months and six months following surgery, every patient came back for a follow-up evaluation. Clinical investigation of the post-grafting healing revealed excellent soft tissue responses to both materials, and no unfavorable side effects were noticed or reported.

In ten individuals, a total of 20 bilateral abnormalities were treated. Ten defect sites received NB + AM treatment, while ten defect sites received NB + BM treatment (Figure 7).

The mean values of PPD, RAL and defect site depth were higher in the pre-operative period compared to the post-operative period. (Table 1, Table 2, Table 3 and Table 4) The PPD, RAL and defect site depth in the post-operative stages in both groups were reduced. For both experimental groups, the preliminary clinical measurements were comparable. 

For sites treated with AM + NB, the average pocket probing depth, relative attachment level, and initial depth of the bone defect were 8.60 mm, 13.70 mm, and 7.76 mm, respectively, while for sites treated with BM + NB, they were 8.7 mm, 14.10 mm, and 7.96 mm. The baseline of both groups exhibited no statistical difference between the sites treated. 

PPD for the AM + NB group was 6.10 1.20 at 3 months and 3.40 0.85 at 6 months, which was statistically significant for each group (*p* < 0.05 for each pair). In contrast, PPD reduction for the BM + NB group was 6.10 1.20 at 3 months and 3.40 0.85 at 6-month intervals.

The RAL for AM + NB was 1.70 ± 0.14 at 3 months and 1.90 ± 0.95 at 6 months while the BM + NB group was 2.90 ± 1.15 at 3 months and 2.70 ± 1.05 at 6 months which was statistically significant (*p* < 0.05) for each pair.

When compared to baseline, both therapy groups considerably improved in terms of defect fill and % bone gain. (Table 3 and Table 4). The average bone fill in AM + NB group was 1.25 ± 0.38 at 6 months and in BM + NB group was 2.64 ± 0.44 at 6 months. The percentage gain for AM + NB was 16% and that for BM + NB was 33%. (Table 4). Statistically significant results were observed in both groups when evaluating all sites at 6 months.

## 4. Discussion

Elective cellular actions that are aided by the use of barrier membranes for tissue exclusion result in the regeneration of the periodontium. Several barrier membrane materials have been utilized in intrabony defects for promoting periodontal regeneration yet to date, none of the materials have been proven as a gold standard. In the present study, the efficacy of GTR therapy using AM and BM barrier membranes was assessed and compared in combination with an allograft NB for the treatment of periodontal intrabony defects. On the evaluation of clinical parameters, both barrier membranes were comparable in terms of pocket depth reduction and gain of RAL at baseline, 3 months and 6 months. These results coincide with those of Jong Jin Suh et al. who assessed the probing depth and CAL at 6 months post operative [10] and Sumegha Srivatsava et al. [11] and Farin Kiany et al. [12] in 2015 who also showed similar results.

The average bone fill was higher in patients treated with BioMesh and nova bone putty than in those treated with AmnioGuard. These results were in conjunction with studies by Jong Jin Suh et al. in 2007, Mopur et al. in 2013 [13], Srivatsava et al. in 2015.

A notable limitation of this study includes the smaller sample size. The duration of the study could have been extended to one year for better observation of results. Because a histological investigation and re-entry procedure were not carried out due to ethical concerns, we were unable to determine if the clinical benefits were brought about by tissue regeneration, repair, and/or a mix of both healing activities. The future of GTR promises to be exciting as new materials in combination with biological activity will become available. It is clear that the “ideal” membrane for use periodontal regenerative therapy is yet to be developed. Based on a graded-biomaterials approach, a biologically active, spatially designed and functionally graded material similar to the natural extracellular matrix could succeed as the next generation of membranes for periodontal regeneration.

## 5. Conclusions

The findings of the current study support the use of bone graft (NB) and barrier membranes like AM/BM for the successful treatment of intrabony defects. The comparison between the two membranes AM and BM resulted in a significant difference in terms of pocket probing depth, amount of defect fill and percentage of bone gain. Both membranes resulted in bone growth of the defect sites but BM yielded a better result (33% bone gain) than AM (16% bone gain).

## Figures and Tables

**Figure 1 ijerph-20-00816-f001:**
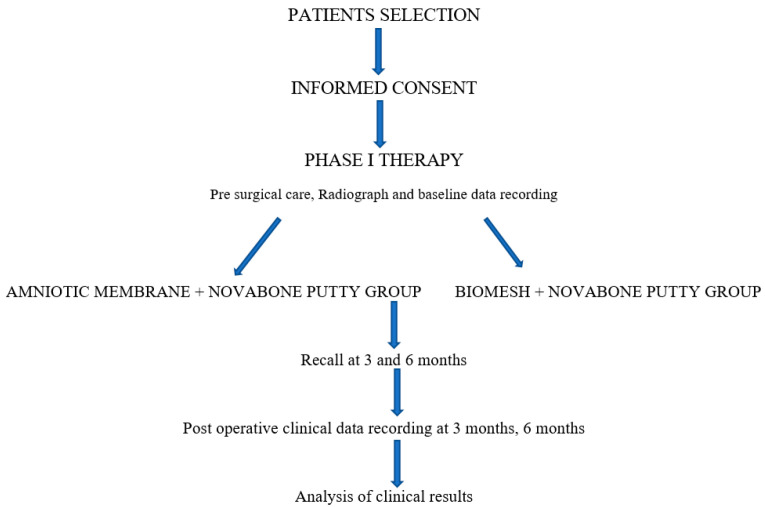
Study protocol.

**Figure 2 ijerph-20-00816-f002:**
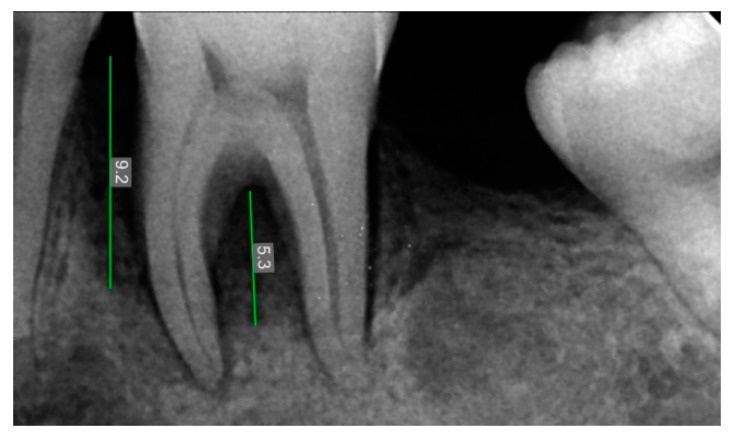
Pre-Operative X-ray of 36 showing bone defect.

**Figure 3 ijerph-20-00816-f003:**
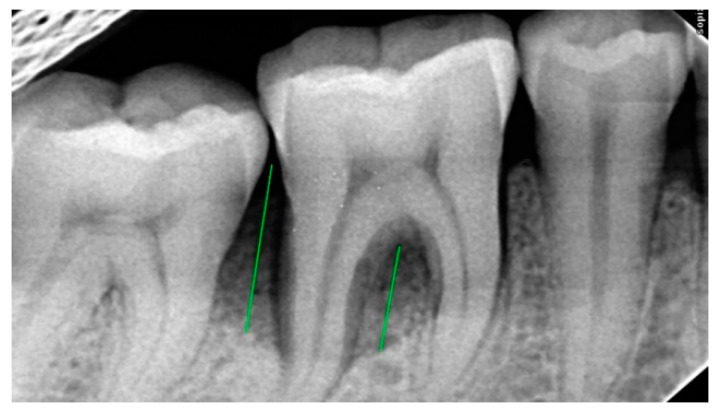
Pre-Operative X-ray of 46 showing bone defect.

**Figure 4 ijerph-20-00816-f004:**
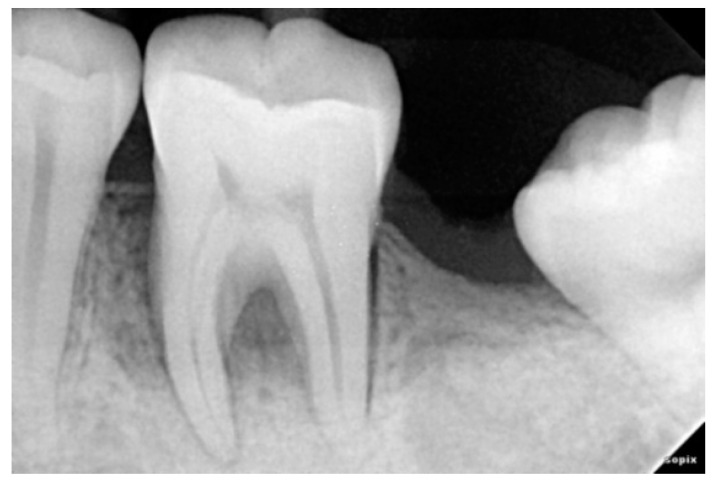
Three Month Post-Operative Radiograph of 36 treated with AM + NB.

**Figure 5 ijerph-20-00816-f005:**
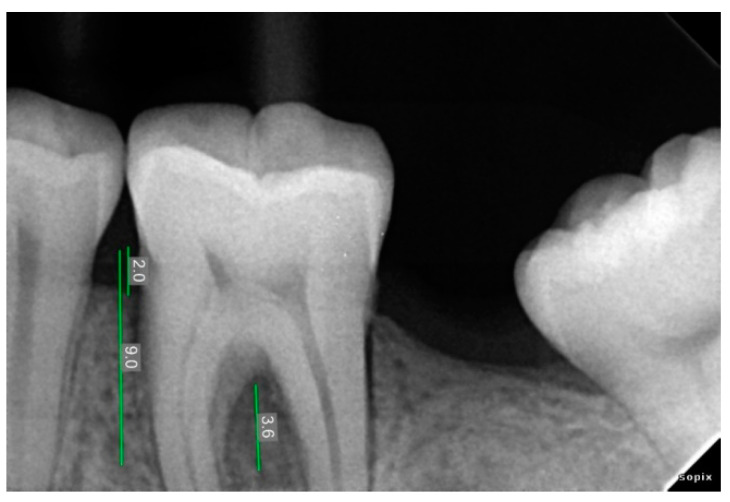
Six Month Post-Operative Radiograph of 36 treated with AM + NB.

**Figure 6 ijerph-20-00816-f006:**
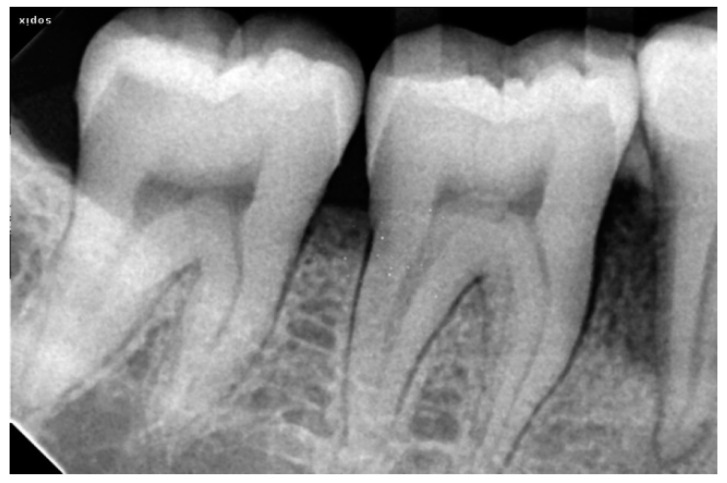
Three Months Post-Operative Radiograph of 46 treated with BM + NB.

**Figure 7 ijerph-20-00816-f007:**
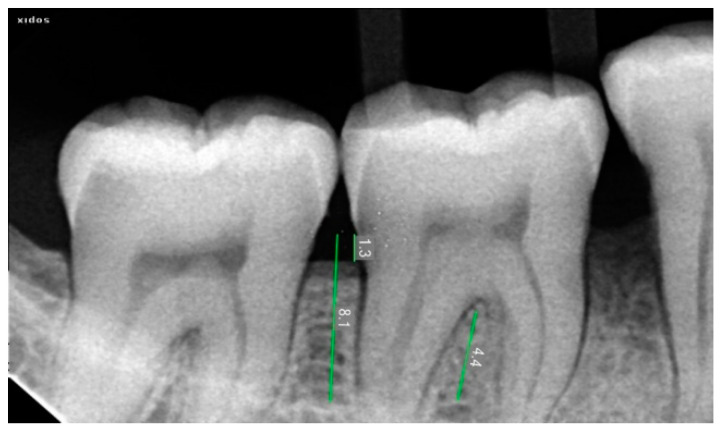
Six Months Post-Operative Radiograph of 46 treated with BM + NB.

**Table 1 ijerph-20-00816-t001:** Pocket Probing Depth (Mean [mm] ± SD) at baseline, 3 months and 6 months (*n* = 10 patients in each group).

	Pre Operative	Post Operative	
	Baseline	3 months	
	(Mean ± SD)	(Mean ± SD)	
AM + NB	8.60 ± 1.17 mm	6.90 ± 1.10 mm	AM + NB
BM + NB	8.70 ± 0.95 mm	6.10 ± 1.20 mm	BM + NB
*p* value	0.853	0.165	*p* value

**Table 2 ijerph-20-00816-t002:** Relative Attachment level (Mean [mm] ± SD) at baseline, 3 months and 6 months (*n* = 10 patients in each group).

	Pre Operative	Post Operative	
	Baseline	3 months	
	(Mean ± SD)	(Mean ± SD)	
AM + NB	13.70 ± 0.95 mm	12.00 ± 0.81 mm	AM + NB
BM + NB	14.10 ± 0.87 mm	11.50 ± 1.35 mm	BM + NB
*p* value	0.393	0.315	*p* value

**Table 3 ijerph-20-00816-t003:** Defect Fill (Mean [mm] ± SD) at baseline, 3 months and 6 months (*n* = 10 patients in each group).

	Defect Fill (Mean ± SD)
AM + NB	1.25 ± 0.38 mm
BM + NB	2.64 ± 0.44 mm
*p* value	0.0005

**Table 4 ijerph-20-00816-t004:** Percentage (%) gain in bone level at 6 months (*n* = 10 patients in each group).

AM + NB	16%
BM + NB	33%

## Data Availability

Not applicable.

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
