# Peer review of "Effect of Amnioguard and Biomesh GTR Membranes with Novabone Putty in the Treatment of Periodontal Osseous Defects—A Communication"

_ijerph, 2023, doi:10.3390/ijerph20010816_

Round 1
Reviewer 1 Report
Title: Some words capitalised and some not. Please double check for consistency.
Abstract: Please change ‘is’ to ‘was’. Please add some values ie. changes in growth etc in the abstract. What is the conclusion? Botha treatments contain Novabone. So what is the real effect? Novabone or vice versa?
Introduction:
Line 4: check typo.
The spelling in the article is NovaBone? Please check for consistency.
‘Aim of The Study’ was written separately from the introduction section. Please check whether this is the right format.
Materials and methods:
Please use the same brand/trade names in the title for NovaBone etc.
Some sentences appear as red.
Figure 1 – not mentioned in the text. What is this figure for?
Figure 4,5, and 6 too. What are these for?
Statistical analysis – please re-phrase the sentence. Mean and standard deviation (SD). Why used Mann Whitney and Friedman test? Please clearly describe this section.
Results:
Table 1 – the table format is not up to the standard for publication. No need to bold the values in the table.
P-value is written as p-value. Should add caption for the table.
Discussion:
Poorly written. Need to discuss about the mechanisms or potential mechanisms of the observed outcomes.
Conclusion:
Poorly written and need to re-write to reflect the content.
Author Response
Comments and Suggestions for Authors
REVIEWER 1:
Title: Some words capitalised and some not. Please double-check for consistency
The title has been edited to sentence case
Abstract: Please change ‘is’ to ‘was’ - Change made
Please add some values ie. changes in growth etc in the abstract. What is the conclusion?
Bone change values added. Both treatments contain Novabone, the membranes were analysed and it was concluded that Biomesh gave better results in our study.
Introduction:
Line 4: check typo - Changes made
The spelling in the article is NovaBone? Discrepancies corrected.
‘Aim of The Study’ was written separately from the introduction section. Please check whether this is the right format - Corrected
Materials and methods:
Please use the same brand/trade names in the title for NovaBone etc - Change made
Some sentences appear as red - Change made
Figure 1Figure 4,5, and 6 too. What are these for?– not mentioned in the text.
The figures have been mentioned in their appropriate paragraphs.
Statistical analysis – please re-phrase the sentence. Mean and standard deviation (SD). Why used Mann Whitney and Friedman test? - Changes made
Results:
Table 1 – the table format is not up to the standard for publication. No need to bold the values in the table.
Changes made
P-value is written as p-value. Should add caption for the table- Change made
Discussion:
Poorly written. Need to discuss about the mechanisms or potential mechanisms of the observed outcomes.
Conclusion:
Poorly written and need to re-write to reflect the content - Change made
REVIEWER 2
Comments and Suggestions for Authors
Statistical analysis is missing - Written as a separate topic
The results and discussion section should be better presented to highlight the most significant and unexpected results and identify correlations, patterns, and relationships among the data, speculations, limitations of work, and deductive arguments. Also, all results should be more integrated with the discussion and should be supported by state-of-the-art studies
Changes made
REVIEWER 3:
Overall
The paper has interesting content, but lacks content on materials and methods, and the intraoral photographs (Figs) are not clear enough, and need to be revised significantly.
Changes made
⑴ Please provide information (e.g., specific composition of the material) and fig on the material (Amnio-Guard ,Biomesh )for comparison. Mentioned in the introduction
(2) Since this is a split-mouse study, please provide information on inclusion criteria and exclusion criteria. Mentioned
⑶ The presence or absence of smoking, age, measurement site, systemic diseases, upper and lower jaw, etc. should be summarized in a table since the number of cases is only 10.
Mentioned
⑷ This is a prospective clinical study on human subjects. A statement regarding ethical review is required
Mentioned

Reviewer 2 Report
The scope of the paper is not clear. Also, the significance and novelty of this work are questionable.
The work contains many flaws, the practical work is not enough for a research paper.
The paper is too short.
The Figures could be combined into two Figures (Panels A, B, C, etc). More experimental work with more Figures should be added.
Statistical analysis is missing.
The discussion is too short and does not interpret the results in terms of the previous state-of-the-art studies:
The results and discussion section should be better presented to highlight the most significant and unexpected results and identify correlations, patterns, and relationships among the data, speculations, limitations of work, and deductive arguments. Also, all results should be more integrated with the discussion and should be supported by state-of-the-art studies.
Author Response

(The authors gave the same response as above.)

Reviewer 3 Report
Overall
The paper has interesting content, but lacks content on materials and methods, and the intraoral photographs (Figs) are not clear enough, and need to be revised significantly.
⑴ Please provide information (e.g., specific composition of the material) and fig on the material (Amnio-Guard ,Biomesh )for comparison.
(2) Since this is a split-mouse study, please provide information on inclusion criteria and exclusion criteria.
⑶ The presence or absence of smoking, age, measurement site, systemic diseases, upper and lower jaw, etc. should be summarized in a table since the number of cases is only 10.
⑷ This is a prospective clinical study on human subjects. A statement regarding ethical review is required
⑸ Photos at the time of surgery are unclear and insufficient, please improve.
Author Response

(The authors gave the same response as above.)

Round 2
Reviewer 2 Report
The paper still poorly written and contains few experimental work to be considered as a research paper. The authors did not respond well to all comments. Also, the discussion is too short and poorly written with no detailed interpretation for data and linking to previous studies.
The newly added tables are not clear and contains flaws. For instance, how come P value is found in the last column and row.
Author Response
2ND ROUND COMMENTS
REVIEWER2
1)The paper still poorly written and contains few experimental work to be considered as a research paper. The authors did not respond well to all comments. Also, the discussion is too short and poorly written with no detailed interpretation for data and linking to previous studies.
There wasn't much data comparing the two products used in this study
2)The newly added tables are not clear and contains flaws. For instance, how come P value is found in the last column and row.
Tables have been changed, the last columns had P value conclusion at 6 months, that has been removed .
REVIEWER 3
- Although appropriately corrected, it does not provide details about AmnioGuard®, NovaBone®, and BioMesh®, the materials being compared in this study. The characteristics of each material should be described in detail, including figures, etc.
We have used minimal photographs and pictures in this study.
Material Characteristics have been added.

Reviewer 3 Report
Although appropriately corrected, it does not provide details about AmnioGuard®, NovaBone®, and BioMesh®, the materials being compared in this study. The characteristics of each material should be described in detail, including figures, etc.
Author Response

(The authors gave the same response as above.)
